# Dense Representation Learning for a Joint-Embedding Predictive Architecture

## Abstract

The joint-embedding predictive architecture (JEPA) recently has shown impressive results in extracting visual representations from unlabeled imagery under a masking strategy. However, we reveal its disadvantage lies in the inadequate grasp of local semantics for dense representations, a shortfall stemming from its masked modeling on the embedding space and the consequent in less discriminative or even missing local semantics. To bridge this gap, we introduce Dense-JEPA, a novel masked modeling objective rooted in JEPA, tailored for enhanced dense representation learning. Our key idea is simple: we consider a set of semantically similar neighboring patches as a target of a masked patch. To be specific, the proposed Dense-JEPA (a) computes feature similarities between each masked patch and its corresponding neighboring patches to select patches having semantically meaningful relations, and (b) employs lightweight cross-attention heads to aggregate features of neighboring patches as the masked targets. Consequently, Dense-JEPA learns better dense representations, which can be beneficial to a wide range of downstream tasks. Through extensive experiments, we demonstrate our effectiveness across various visual benchmarks, including ImageNet-1K image classification, ADE20K semantic segmentation, and COCO object detection tasks.

## 1 Introduction

The success of self-supervised learning (SSL) frameworks (Chen et al., 2020a; Chen & He, 2021; He et al., 2020; Grill et al., 2020), especially in harnessing vast reservoirs of unlabeled images, has been undeniable in the computer vision community. Model architectures like the Vision Transformer (ViT; Dosovitskiy et al. (2021)) have consistently garnered significant attention, and initial attempts at seamless integration with SSL have indeed demonstrated potential (Chen et al., 2021; Xie et al., 2021; Caron et al., 2021). In particular, Masked autoencoder (MAE; He et al. (2021)), which reconstructs missing patches on pixel space, has achieved advanced success in various visual downstream tasks, such as image classification, object detection, and semantic segmentation.

Recently, the image-based joint-embedding predictive architecture (I-JEPA; Assran et al. (2023)) has shown promising results in learning self-supervised representations by leveraging a masking strategy to reconstruct representations of masked patches. Specifically, I-JEPA uses a masked image to predict the representations of various unmasked blocks located in the same image. Nevertheless, we observed that this approach often produces inaccurate self-attention maps, as illustrated in Figure 1). These inaccuracies can be attributed to the inability to capture a deep understanding of local semantics, which is essential for dense prediction tasks.

One challenge posed by existing SSL approaches built upon ViTs is the potential lack of local semantics in the extracted representations from disjoint input patches, where local semantics are naturally intertwined within the image patches. This inspires us to incorporate explicit processing of these local semantics in I-JEPA for improved dense representation learning. Our key idea is to generate a semantically meaningful target, encompassing local semantics, for each masked patch by harnessing similarities among patches within a neighborhood. During pre-training, we employ the self-supervised dense target to capture local semantics, which can be advantageous for a variety of dense prediction tasks, such as object detection and semantic segmentation.

In this paper, we introduce Dense-JEPA, a novel dense representation learning framework with a Joint-Embedding Predictive Architecture. Our goal is to generate dense targets capturing local se-

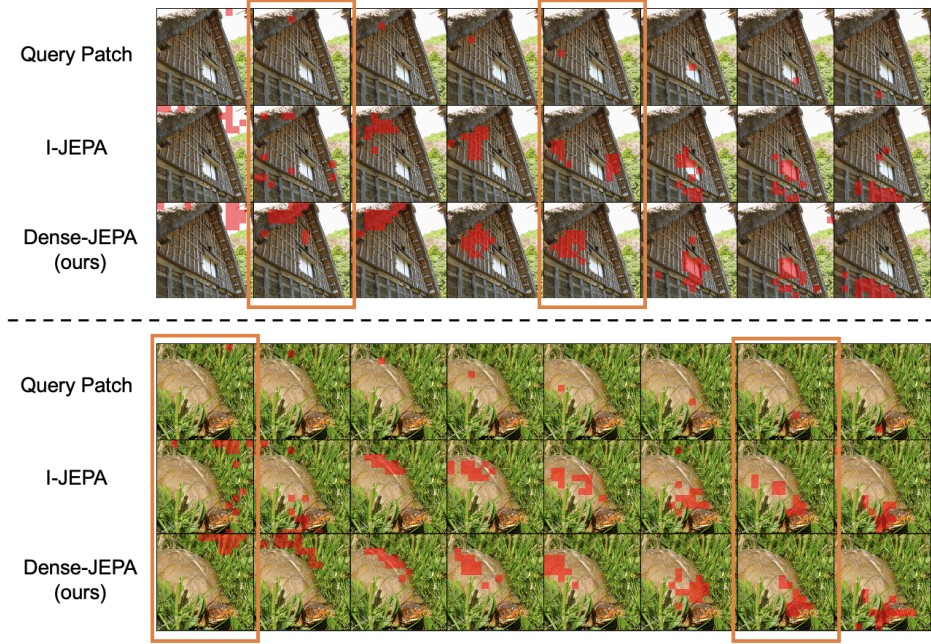

Figure 1: **Visualizations for qualitative comparison.** We visualize the top-10% of highly correlated patches by thresholding the attention maps of query patches in the last layer using pre-trained I-JEPA and our method. The patches extracted by I-JEPA are separated and do not specifically focus on the location of the given query patch, especially for examples from the boxes highlighted in yellow. In contrast, Dense-JEPA performs better by encouraging the model to learn local semantics.

mantics that can serve as an alternative masked modeling objective, which can be incorporated with the joint-embedding predictive architecture (LeCun, 2022). To this end, we propose Masked Semantic Neighboring to find semantically similar neighboring patches for masked patches and Local Aggregation Target to generate the dense targets from them. Specifically, Masked Semantic Neighboring computes feature similarities between each masked patch and its corresponding neighboring patches to select semantically similar patches, and Local Aggregation Target employs lightweight cross-attention heads to aggregate features of chosen neighboring patches as the dense targets for masked patches. Consequently, the proposed Dense-JEPA learns better dense representations, as shown in Figure 1, which can be beneficial to dense prediction downstream tasks such as semantic segmentation, object detection, and local prediction tasks.

To demonstrate the effectiveness of Dense-JEPA, we conduct extensive downstream experiments after pre-training ViT-B/16 and ViT-L/16 (Dosovitskiy et al., 2021) on ImageNet-1k; these experiments include ImageNet-1K image classification, COCO object detection, ADE20K semantic segmentation, DAVIS video segmentation, and Clevr local prediction benchmarks. Our experimental results demonstrate that the Dense-JEPA improves the performance of I-JEPA with a large margin and even outperforms other SSL baselines (He et al., 2021; Chen et al., 2021; Bao et al., 2021) on various benchmarks; for example, our method achieved +1.4 mIoU (*i.e.,* $47.6 \rightarrow 49.0$) on ADE20K semantic segmentation, +1.7 $(\mathcal{J}\&\mathcal{F})_m$ (*i.e.,* $56.6 \rightarrow 58.3$) on DAVIS video segmentation, +1.0 $AP^{box}$ (*i.e.,* $49.9 \rightarrow 50.9$) on COCO object detection, and +1.1 $AP^{mask}$ (*i.e.,* $44.5 \rightarrow 45.6$) on COCO instance segmentation. Furthermore, we observed that Dense-JEPA also can improve linear evaluation performances (*i.e.,* $77.5 \rightarrow 78.2$) of I-JEPA on ImageNet-1k benchmark, which demonstrates that the proposed method is not only beneficial to dense representation learning but also can enhance the quality of global image representations.

Overall, our work highlights the effectiveness of considering a dense target for masked modeling on the embedding space, and we hope that our work could inspire researchers to further explore dense representation learning in a self-supervised manner.

## 2 RELATED WORK

**Self-supervised Visual Representation Learning.** Self-supervised visual learning aims to mine the internal characteristics of images without annotations by applying well-designed pretext tasks. Early non-transformer researchers introduced instance-level (Wu et al., 2018; Chen et al., 2020a;b; Grill et al., 2020; He et al., 2020; Chen et al., 2020c; Chen & He, 2021; Zbontar et al., 2021) and cluster-based (Caron et al., 2020; Li et al., 2021a; Wang et al., 2021; Mo et al., 2021; 2022) contrastive learning to pull representations from positive samples closer while pushing away features from negative pairs. Recently, contrastive learning has been widely used in self-supervised vision transformers (Chen et al., 2021; Xie et al., 2021; Caron et al., 2021; Yun et al., 2022; Mo et al., 2023) to achieve promising performance on visual downstream tasks. Typically, MoCo-v3 (Chen et al., 2021) introduced a momentum encoder in ViT (Dosovitskiy et al., 2021) to minimize the distance between representations of two augmented views from the base encoder and momentum one. To capture the local-to-global alignment, DINO (Caron et al., 2021) used a momentum encoder with multi-crop training to achieve knowledge distillation in the vision transformer.

**Masked Image Modeling.** Masked image modeling has been extensively explored in previous literature (Bao et al., 2021; Atito et al., 2021; He et al., 2021; Wei et al., 2022; Xie et al., 2022) to reconstruct the masked image patch given the unmasked counterpart as clues. Early approaches (Bao et al., 2021; Atito et al., 2021; He et al., 2021; Li et al., 2021b; Shi et al., 2022) designed customized masking strategies (*e.g.*, random, block-wise) as pre-text tasks during pre-training. For example, block-wise masking was introduced in BEiT (Bao et al., 2021) to learn transferrable visual representations by recovering discrete tokens of masked image patches. Given features extracted from the 25% unmasked patches, the seminal work, MAE (He et al., 2021) directly reconstructed missing pixels of 75% masked patches, and showed promising performances on various downstream tasks.

**Joint-Embedding Predictive Architectures.** Joint-Embedding Predictive Architectures (JEPA; Le-Cun (2022)) learn to predict the embeddings of a signal from a compatible signal using a predictor conditioned on a latent variable to achieve prediction. Based on JEPA, the image-based JEPA, I-JEPA (Assran et al., 2023), recently proposed to predict multiple target block representations given the output of the context encoder under a masking strategy. The key characteristic of I-JEPA is that its objective is located on the embedding space, while masked image modeling models do on the pixel (or token) space. Despite it enabling efficient pre-training of I-JEPA, however, the masked modeling target on the embedding space could pose degradation in dense representation learning if the target representations were discriminative or lack local semantics among themselves. In contrast, we aim to develop a novel masked modeling objective incorporated with JEPA to generate target representations capturing local semantics for learning better dense representations.

## 3 METHOD

In this section, we present a novel masked modeling framework, coined Dense-JEPA, designed for the joint-embedding predictive architecture to enhance understanding local semantics within images. Our key idea is that semantically similar representations can provide local semantics as a masked modeling objective by enforcing them to have similar representations. We first provide preliminaries in Section 3.1 and then present details of two modules, Masked Semantic Neighboring in Section 3.2 and Local Aggregation Target in Section 3.3. Figure 2 shows an overall illustration of the proposed method, Dense-JEPA.

### 3.1 PRELIMINARIES

We first describe the problem setup and notations and then revisit the Image-based Joint-Embedding Predictive Architectures (I-JEPA; Assran et al. (2023)), which is a self-supervised visual representation learning under masked modeling.

**Problem Setup and Notations.** Given an image with a dimension of $3 \times H \times W$ and a patch resolution of $P$, our goal is to learn a masked autoencoder framework with an encoder $f_e(\cdot)$ and a decoder $f_d(\cdot)$ to recover the masked patches using unmasked ones. We formally denote patch embeddings of raw input via each linear projection layer, *i.e.*, $\mathbf{x} \in \mathbb{R}^{N \times D}$, $H$ and $W$ are the height

and width of each image, and $D$ is the dimension of features. Note that $N = H/P \times W/P$ and $N$ is the total number of patches.

**Masked Autoencoder.** To address the masked image modeling problem, MAE (He et al., 2021) first applied a random masking set $M$ along the total number of patches, and then an encoder to extract features from unmasked patches. Finally, unmasked embeddings and masked tokens were concatenated into a decoder to recover the raw pixels of masked patches. The vanilla masking loss for each image is calculated with the mean square loss between the targeted $\mathbf{p}_i$ and predicted normalized pixels $\hat{\mathbf{p}}_i$ as:

$$\mathcal{L}_{\text{MAE}} = \frac{1}{|M|} \sum_{i \in M} ||\mathbf{p}_i - \hat{\mathbf{p}}_i||_2^2, \tag{1}$$

where $|M|$ denotes the total number of masked patches in the masking set $M$.

**Image-based Joint-Embedding Predictive Architecture.** To tackle the masked image modeling task, I-JEPA (Assran et al., 2023) introduced a context encoder $f_\theta(\cdot)$, a target encoder $f_{\bar{\theta}}(\cdot)$, and a predictor $g_\theta(\cdot)$, to predict the $M$ target block representations $\mathbf{s}_y(1), ..., \mathbf{s}_y(M)$ given the output of the context encoder, $\mathbf{s}_x$, For a target block $\mathbf{s}_{y_i}$ corresponding to a target mask $\mathcal{B}_i$, the predictor $g_\theta(\cdot, \cdot)$ takes as input the output of the context encoder $\mathbf{s}_x$ and a mask token for each patch to predict $\{\mathbf{m}_j\}_{j \in \mathcal{B}_i}$, and outputs the patch-level prediction $\{\hat{\mathbf{s}}_{y_j}\}_{j \in \mathcal{B}_i}$, that is, $\{\hat{\mathbf{s}}_{y_j}\}_{j \in \mathcal{B}_i} = g_\theta(\mathbf{s}_x, \{\mathbf{m}_j\}_{j \in \mathcal{B}_i})$ The masking objective is optimized by the average $L_2$ distance between the predicted patch-level representations $\hat{\mathbf{s}}_{y_j}$ and the target patch-level representation $\mathbf{s}_{y_j}$, which is formulated as:

$$\mathcal{L}_{\text{I-JEPA}} = \frac{1}{|M|} \sum_{i=1}^{M} \sum_{j \in \mathcal{B}_i} ||\mathbf{s}_{y_j} - \hat{\mathbf{s}}_{y_j}||_2^2, \tag{2}$$

where $|M|$ denotes the total number of target blocks, and $\mathcal{B}_i$ is the mask corresponding to the $i$-th target block.

### 3.2 DENSE-JEPA: MASKED SEMANTIC NEIGHBORING

However, a masked modeling target in the representation space like I-JEPA could pose a challenge in terms of missing local semantics if the target patch-level representations $\mathbf{s}_{y_j}$ were less discriminative among themselves. As shown in Figure 1, we also observed that I-JEPA often generates inaccurate self-attention maps, and it arguably indicates its deficiency in comprehending local semantics. To tackle this, we aim to generate target representations capturing local semantics that can serve as an alternative masked modeling objective, which can be incorporated with the joint-embedding predictive architecture (LeCun, 2022). We note that prior investigation on patch-level representation learning (Yun et al., 2022) inspires us to explore similarities among patch-level representations located in a neighborhood. To this end, we propose Masked Semantic Neighboring module to find semantically similar neighboring patches for masked patches and Local Aggregation Target module (see Section 3.3) to make them have similar target representations.

**Masked Semantic Neighboring.** For patches in a given masked block, we aim to find their neighboring patches semantically similar, as neighboring patches often share a semantic context. In order to sample semantically similar patches from the neighborhood $\mathcal{N}_i$, we compute the dense semantic similarity $d(i, j)$ between the query patch $\mathbf{x}_i$ and its neighboring patch $\mathbf{x}_j$ for all $j \in \mathcal{N}_i$ based on representations from the target encoder $f_{\bar{\theta}}(\cdot)$, which is formulated as:

$$d(i, j) = \frac{f_{\bar{\theta}}(\mathbf{x}_i)^\top f_{\bar{\theta}}(\mathbf{x}_j)}{||f_{\bar{\theta}}(\mathbf{x}_i)||_2 ||f_{\bar{\theta}}(\mathbf{x}_j)||_2}, \tag{3}$$

where $f_{\bar{\theta}}(\mathbf{x}_i), f_{\bar{\theta}}(\mathbf{x}_j) \in \mathbb{R}^{1 \times D}$, and $|| \cdot ||_2$ denotes the $\ell_2$-norm operator. With the computed similarity scores, we apply a ranking on the neighboring patches $\{\mathbf{x}_j\}_{j \in \mathcal{N}_i}$ and select a set of dense patches $\{\mathbf{x}_j\}_{j \in \mathcal{P}_i}$ with top-$k$ highest similarities, where $\mathcal{P}_i$ denotes a set of dense patch indices in the neighborhood, and $k$ is the number of dense patches, *i.e.*, $k = |\mathcal{P}_i|$. Unless stated otherwise, we use $k = 4$ for our experiments.

### 3.3 DENSE-JEPA: LOCAL AGGREGATION TARGET

We remark that our goal is to generate target representations capturing local semantics and discriminative among themselves. With the benefit of the selected neighboring patches having similar

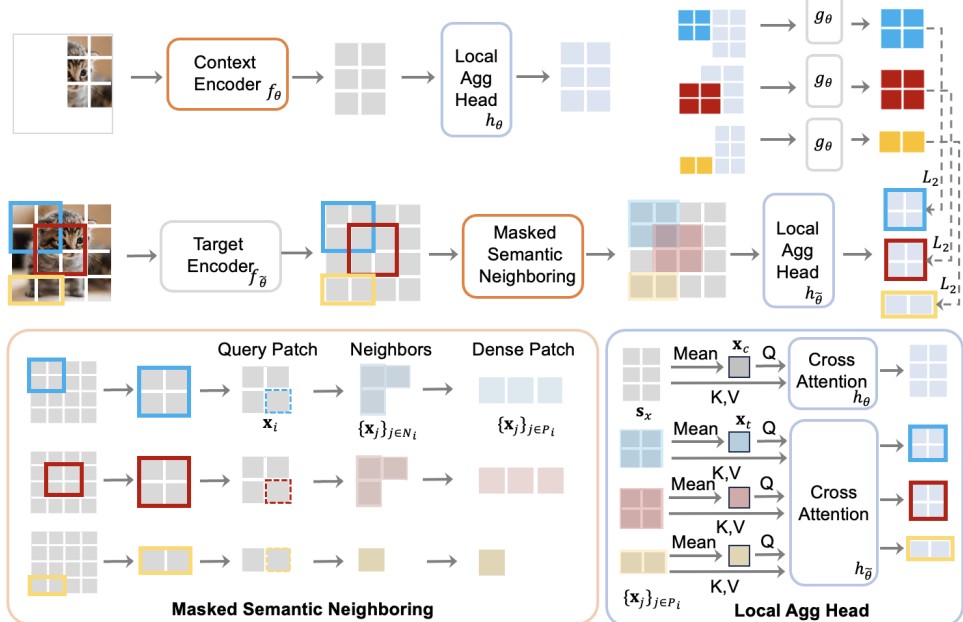

Figure 2: Illustration of the proposed novel masked modeling framework (Dense-JEPA), rooted in I-JEPA, tailored for enhanced dense representation learning. The Masked Semantic Neighboring module computes the dense semantic similarity between the query patch and its neighboring patch based on representations from the target encoder $f_{\tilde{\theta}}(\cdot)$ to select semantically similar patches from the neighborhood. Then Local Aggregation Target module composed of a context patch aggregation head $h_\theta(\cdot)$ and a target patch aggregation head $h_{\tilde{\theta}}(\cdot)$, aggregates target features of selected patches using cross-attention to construct dense targets. Finally, the model is optimized by the average $L_2$ distance between the predicted dense representations and the target dense representation.

semantics, we introduce Local Aggregation Target module composed of a context patch aggregation head $h_\theta(\cdot)$ and a target patch aggregation head $h_{\tilde{\theta}}(\cdot)$. Specifically, we aggregate target representations of selected patches $\{f_{\tilde{\theta}}(\mathbf{x}_j)\}_{j \in \mathcal{P}_i}$ using cross-attention head $h_\theta(\cdot)$ to construct dense target $\mathbf{s}_i^{\text{LAT}}$ for enforcing semantically similar patches could have the similar dense targets. Simultaneously, we symmetrically apply context aggregation head $h_\theta(\cdot)$ to produce corresponding context $\mathbf{s}_x^{\text{LAT}}$ from context representations of unmasked patches as:

$$\mathbf{s}_i^{\text{LAT}} = h_{\tilde{\theta}}(\{\mathbf{x}_j\}_{j \in \mathcal{P}_i}, \mathbf{x}_t), \quad \mathbf{s}_x^{\text{LAT}} = h_\theta(\mathbf{s}_x, \mathbf{x}_c), \tag{4}$$

where $\mathbf{s}_x$ denotes context embeddings and $\mathbf{x}_t, \mathbf{x}_c$ denote the averaged embeddings from all patches in the target encoder and only unmasked patches in the context encoder, respectively. The cross-attention operator $h_\theta(\cdot)$ and $h_{\tilde{\theta}}(\cdot)$ is formulated as:

$$h_{\tilde{\theta}}(\{\mathbf{x}_j\}_{j \in \mathcal{P}_i}, \mathbf{x}_t) = \texttt{Softmax}\left(\frac{\mathbf{x}_t\{\mathbf{x}_j\}_{j \in \mathcal{P}_i}^\top}{\sqrt{D}}\right)\{\mathbf{x}_j\}_{j \in \mathcal{P}_i},$$

$$h_\theta(\mathbf{s}_x, \mathbf{x}_c) = \texttt{Softmax}\left(\frac{\mathbf{x}_c\mathbf{s}_x^\top}{\sqrt{D}}\right)\mathbf{s}_x, \tag{5}$$

where $D$ is the dimension of embeddings. For a given target block $\mathbf{s}_{y_i}^{\text{LAT}}$ corresponding to a target mask $\mathcal{B}_i$, the predictor $g_\theta(\cdot, \cdot)$ takes as input the output of the context patch aggregation head $\mathbf{s}_x^a$ and a mask token for each patch to predict $\{\mathbf{m}_j\}_{j \in \mathcal{B}_i}$, and outputs a dense prediction $\{\hat{\mathbf{s}}_{y_j}^{\text{LAT}}\}_{j \in \mathcal{B}_i} = g_\theta(\mathbf{s}_x^{\text{LAT}}, \{\mathbf{m}_j\}_{j \in \mathcal{B}_i})$. The new masking objective is optimized by the average $L_2$ distance between the predicted dense representations $\hat{\mathbf{s}}_{y_j}^{\text{LAT}}$ and the target dense representation $\mathbf{s}_i$ in the $i$-th block, which is formulated as:

$$\mathcal{L}_{\texttt{Dense-JEPA}} = \frac{1}{|M|}\sum_{i=1}^{M}\sum_{j \in \mathcal{B}_i}||\mathbf{s}_i^{\text{LAT}} - \hat{\mathbf{s}}_{y_j}^{\text{LAT}}||_2^2, \tag{6}$$

Table 1: **ADE20K semantic segmentation, COCO object detection, and instance segmentation.** We fine-tuned pre-trained ViT-B/16 models to perform ADE20K semantic segmentation and COCO object detection and instance segmentation. The mIoU, $AP^{box}$, and $AP^{mask}$ metrics denote the results of ADE20K segmentation, COCO detection, and COCO segmentation, respectively. The best results are indicated in **bold**.

| Method | Pre-train data | mIoU | $AP^{box}$ | $AP^{mask}$ |
|---|---|---|---|---|
| Supervised | ImageNet-1K w/ labels | 47.4 | 47.9 | 42.9 |
| DINO (Caron et al., 2021) | ImageNet-1K | 46.8 | 50.1 | 43.4 |
| MoCo v3 (Chen et al., 2021) | ImageNet-1K | 47.3 | 47.9 | 42.7 |
| BEiT (Bao et al., 2021) | ImageNet-1K+DALLE | 47.1 | 49.8 | 44.4 |
| MAE (He et al., 2021) | ImageNet-1K | 48.1 | 50.3 | 44.9 |
| I-JEPA (Assran et al., 2023) | ImageNet-1K | 47.6 | 49.9 | 44.5 |
| Dense-JEPA (ours) | ImageNet-1K | **49.0** | **50.9** | **45.6** |

where $|M|$ denotes the total number of target blocks, and $\mathcal{B}_i$ is the target mask corresponding to the $i$-th target block. In terms of semantic similarity among patches, the closer the final target representations, pre-training through these targets would promote the enhancement of learned embeddings that encompass local semantics.

## 4 EXPERIMENTS

### 4.1 EXPERIMENTAL SETUP

**Datasets.** Following previous methods (He et al., 2021; Assran et al., 2023), we use ImageNet-1K (Deng et al., 2009) for image classification, MS-COCO (Lin et al., 2014) for object detection and instance segmentation, and ADE20K (Zhou et al., 2017; 2018) for semantic segmentation. We closely follow previous work (Chen et al., 2021; Xie et al., 2021; Caron et al., 2021), and adopt the Mask R-CNN (He et al., 2017) as the detector. The ViT-Base (Dosovitskiy et al., 2021) backbone weights are initialized with weights pre-trained on ImageNet-1K using our Dense-JEPA. Following the settings in (He et al., 2021; Bao et al., 2021), we use the UPerNet approach (Xiao et al., 2018) based on our ImageNet-1K pre-trained ViT-Base for evaluation. For a fair comparison, we fine-tune the detector with the same learning rate in (He et al., 2021; Bao et al., 2021). For video object segmentation, we use DAVIS-2017 dataset containing 60 training, 30 validation, and 60 testing videos. For local prediction tasks on Clevr (Johnson et al., 2016), we follow the previous work (Assran et al., 2023) and use Clevr/Count and Clevr/Dist.

**Evaluation Metrics.** We follow previous masked image modeling work (He et al., 2021; Bao et al., 2021) to report the classification accuracy of linear probing and fine-tuning. For object detection and instance segmentation on MS-COCO, we apply $AP^{box}$ and $AP^{mask}$ as metrics for the bounding boxes and the instance masks. mIoU results are reported to evaluate semantic segmentation on ADE20K. For video object segmentation on DAVIS-2017, we use Jabri-based $(\mathcal{J}\&\mathcal{F})_m$, $\mathcal{J}_m$, $\mathcal{F}_m$ as metrics to evaluate the quality of frozen representations of image patches by segmenting scenes with the nearest neighbor between consecutive frames. For local prediction tasks on Clevr, we use object counting and depth prediction to evaluate the linear probing performance of our model.

**Implementation.** For input images, the resolution is resized to $224 \times 224$, *i.e.*, $H = W = 224$. We follow prior work (He et al., 2021; Assran et al., 2023) and apply a patch size of 16, *i.e.*, $P = 16$. The small, base, and large models of ViT (Dosovitskiy et al., 2021) architecture are used for experiments. We set the embedding dimension of the predictor to 384, and keep the number of self-attention heads the same as the backbone context-encoder. For the smaller ViT-S/16 and ViT-B/16 context-encoder, we set the depth of the predictor as 6. For ViT-L/16 context-encoders, we set the depth of the predictor to 12. Following I-JEPA (Assran et al., 2023), we use AdamW to optimize the context-encoder and predictor weights. We train our model using the default batch size of 2048, and the learning rate linearly increased from 1e-4 to 1e-3 during the first 15 epochs of pre-training, and decay to 1e-6 following a cosine schedule. The weight decay is linearly increased from 0.04 to 0.4, and the target-encoder weights are initialized the same as the context-encoder weights, and updated via an exponential moving average. We use a momentum value of 0.996, and linearly increase this

Table 2: **DAVIS video object segmentation.** We perform DAVIS 2017 video object segmentation using ImageNet-1K pre-trained ViT-B/16 and ViT-L/16 models. We report Jabri-based metrics $(\mathcal{J}\&\mathcal{F})_m$, $\mathcal{J}_m$, $\mathcal{F}_m$ to evaluate the quality of pre-trained representations. The best results are indicated in **bold**.

| Method | Backbone | $(\mathcal{J}\&\mathcal{F})_m$ | $\mathcal{J}_m$ | $\mathcal{F}_m$ |
|---|---|---|---|---|
| MAE (He et al., 2021) | ViT-B/16 | 51.0 | 49.4 | 52.6 |
| | ViT-L/16 | 53.4 | 52.5 | 54.3 |
| I-JEPA (Assran et al., 2023) | ViT-B/16 | 56.2 | 56.1 | 56.3 |
| | ViT-L/16 | 56.6 | 56.3 | 56.9 |
| Dense-JEPA (ours) | ViT-B/16 | **57.7** | **56.7** | **58.7** |
| | ViT-L/16 | **58.3** | **57.3** | **59.2** |

Table 3: **ImageNet-1K image classification.** We perform a linear evaluation on pre-trained ViT-B/16 and ViT-L/16 models for image classification on ImageNet-1K benchmark. We report the top-1 accuracy to evaluate the quality of pre-trained representations. The best results are indicated in **bold**.

| Method | Backbone | Epochs | Top-1 Accuracy |
|---|---|---|---|
| data2vec (Baevski et al., 2022) | ViT-L/16 | 1600 | 77.3 |
| MAE (He et al., 2021) | ViT-B/16 | 1600 | 68.0 |
| | ViT-L/16 | 1600 | 76.0 |
| I-JEPA (Assran et al., 2023) | ViT-B/16 | 600 | 72.9 |
| | ViT-L/16 | 600 | 77.5 |
| Dense-JEPA (ours) | ViT-B/16 | 600 | **73.8** |
| | ViT-L/16 | 600 | **78.2** |

value to 1.0. For masking, we use the same strategy and settings as I-JEPA (Assran et al., 2023) for 4 possibly overlapping target blocks masks. Our small, base, and large models are pre-trained on ImageNet-1K (Deng et al., 2009) for 600 epochs.

## 4.2 COMPARISON TO PRIOR WORK

In this work, we propose a novel and effective framework for dense representation learning with a joint-embedding predictive architecture. In order to demonstrate the effectiveness of the proposed Dense-JEPA, we comprehensively compare it to previous mask image modeling baselines (He et al., 2021; Baevski et al., 2022; Chen et al., 2022; Assran et al., 2023).

**Detection and Segmentation tasks.** For the ADE20K semantic segmentation and COCO object detection & instance segmentation benchmarks, we report the quantitative comparison results in Table 1; our method achieved the best results regarding all the metrics compared to previous mask modeling baselines. In particular, the proposed Dense-JEPA outperforms I-JEPA (Assran et al., 2023), the current image-based joint-embedding predictive architecture by 0.9@mIoU. Also, we achieve significant performance gains of 1.0@AP$^{\texttt{box}}$ and 1.1@AP$^{\texttt{mask}}$ on COCO object detection and instance segmentation compared to I-JEPA, which indicates the importance of leveraging the self-supervised dense targets to capture local semantics for dense prediction tasks. Furthermore, we observed that Dense-JEPA even can achieve better results than the strongest baseline, MAE (He et al., 2021), a generative autoencoder architecture for masked image modeling.

In addition, our method shows significant and consistent gains in the DAVIS 2017 video object segmentation benchmark as shown in Table 2. Compared to I-JEPA, ours achieved the results gains of $1.5@(\mathcal{J}\&\mathcal{F})_m$, $0.6@\mathcal{J}_m$, and $2.4@\mathcal{F}_m$ on ViT-B/16. Moreover, the margins increased more significantly when we evaluated the large-scale backbone, ViT-L/16, by $1.7@(\mathcal{J}\&\mathcal{F})_m$, which shows a scaling behavior of ours. Overall, these significant improvements reported in Tables 1 and 2 demonstrate the superiority of our approach to learning local semantics during self-supervised pre-training.

**Image classification task.** Here, we validate the quality of our learned global representation by performing the common linear evaluation task on ImageNet-1k. Table 3 summarizes the results; Dense-JEPA outperforms all the baselines in Table 3. For example, Dense-JEPA achieved 78.2%

Table 4: **Clever object counting and depth prediction.** We perform a linear evaluation on pre-trained models for Clever object counting and depth prediction benchmarks. The Clevr/Count and Clevr/Dist metrics denote the result of object counting and depth prediction tasks, respectively. The best results are indicated in **bold**, and the second best ones are underlined.

| Method | Backbone | Clevr/Count | Clevr/Dist |
|---|---|---|---|
| DINO (Caron et al., 2021) | ViT-B/8 | 86.6 | 53.4 |
| iBOT (Zhou et al., 2022) | ViT-L/16 | 85.7 | 62.8 |
| data2vec (Baevski et al., 2022) | ViT-L/16 | 85.3 | 71.3 |
| MAE (He et al., 2021) | ViT-B/16 | **86.6** | **70.8** |
| | ViT-L/16 | **92.1** | **73.0** |
| I-JEPA (Assran et al., 2023) | ViT-B/16 | 82.2 | 70.7 |
| | ViT-L/16 | 85.6 | 71.2 |
| Dense-JEPA (ours) | ViT-B/16 | 83.5 | 71.1 |
| | ViT-L/16 | 87.1 | 71.8 |

Table 5: **Ablation studies on component analysis.** We perform ablation studies on Masked Semantic Neighboring (MSN) and Local Aggregation Target (LAT) modules using a pre-trained ViT-S/16 model on the DAVIS benchmark. The best results are indicated in **bold**.

| MSN | LAT | $(\mathcal{J}\&\mathcal{F})_m$ | $\mathcal{J}_m$ | $\mathcal{F}_m$ |
|---|---|---|---|---|
| ✗ | ✗ | 53.7 | 52.5 | 54.8 |
| ✓ | ✗ | 55.2 | 54.3 | 56.1 |
| ✗ | ✓ | 54.6 | 53.3 | 55.9 |
| ✓ | ✓ | **57.1** | **55.7** | **58.5** |

top-1 accuracy on ViT-L/16, while MAE and I-JEPA did 76.0% and 77.5%, respectively. These results further indicate the benefit of the proposed method in learning the global semantics of images.

**Other low-level tasks.** We also present additional Clevr benchmarks for measuring abilities of object-counting and depth prediction. Table 4 shows linear evaluation results of Dense-JEPA on the Clevr counting and depth benchmarks. Compared to I-JEPA (Assran et al., 2023), we achieve the results gains of 1.3@Clevr/Count and 0.4@Clevr/Dist using ViT-B/16. Moreover, we observe similar scaling behavior as shown in Table 2, with increased improvements on ViT-L/16. These results where MAE achieved the best scores arguably show that JEPA objectives on the embedding space would have the potential to pose degradation in dense representation learning.

### 4.3 EXPERIMENTAL ANALYSIS

In this section, we performed ablation studies to demonstrate the benefit of the proposed Masked Semantic Neighboring and Local Aggregation Target modules. Here, we conducted extensive experiments on ImageNet-1k pre-trained ViT-S/16 to explore the impact of neighbors and dense pairs, types of local aggregation heads, and learned meaningful patch-level representations.

**Masked Semantic Neighboring & Local Aggregation Target.** In order to demonstrate the effectiveness of Masked Semantic Neighboring (MSN) and Local Aggregation Target (LAT), we ablate the impact of each module and report the quantitative results on DAVIS 2017 video object segmentation benchmark with ViT-S/16 in Table 7. As shown in the table, adding MSN to the vanilla baseline highly increases the results of $1.5@(\mathcal{J}\&\mathcal{F})_m$, which validates the benefit of masked semantic neighboring in finding semantically similar neighboring patches for masked patches. Meanwhile, introducing only LAT in the baseline increases the video segmentation performance regarding all metrics. More importantly, incorporating MSN and LAT into the baseline significantly raises the performance by $3.4@(\mathcal{J}\&\mathcal{F})_m$. These improving results validate the importance of MSN and LAT in extracting local semantics from self-supervised dense targets for better representations.

**Impact of Neighbors and Dense Pairs.** To explore the impact of neighbors in neighboring and the number of selected dense pairs, we ablated the neighbors from $\{3 \times 3, 5 \times 5, \text{All patches}\}$ and varied the number of dense pairs from $\{1, 2, 4, 8\}$. The quantitative results on the DAVIS benchmark with ViT-S/16 are reported in Table 6. As shown in the table, the proposed Dense-JEPA achieved the

Table 6: **Ablation studies on hyperparameters.** We perform ablation studies using a pre-trained ViT-S/16 model to explore effects on neighbors and dense pairs in the Masked Semantic Neighboring module. The best results are indicated in **bold**.

| Neighbors | # Dense Pairs | $(\mathcal{J}\&\mathcal{F})_m$ | $\mathcal{J}_m$ | $\mathcal{F}_m$ |
|---|---|---|---|---|
| 3×3 | | **57.1** | **55.7** | **58.5** |
| 5×5 | 4 | 56.3 | 54.9 | 57.7 |
| All patches | | 48.5 | 46.7 | 50.3 |
| | 1 | 55.9 | 54.3 | 57.5 |
| 3×3 | 2 | 56.3 | 54.9 | 57.7 |
| | 8 | 56.7 | 55.3 | 58.1 |

Table 7: **Ablation studies on aggregation head.** We perform ablation studies on context and target aggregation heads in Local Aggregation Target modules using two types (Cross-attention & Self-attention). The best results are indicated in **bold**.

| Context Head $h_\theta$ | Target Head $h_{\tilde{\theta}}$ | $(\mathcal{J}\&\mathcal{F})_m$ | $\mathcal{J}_m$ | $\mathcal{F}_m$ |
|---|---|---|---|---|
| Cross-attention | Cross-attention | **57.1** | **55.7** | **58.5** |
| Self-attention | Cross-attention | 53.7 | 52.5 | 54.9 |
| Self-attention | Self-attention | 50.6 | 49.5 | 51.7 |

best performance of $(\mathcal{J}\&\mathcal{F})_m$ when we use $3 \times 3$ neighbors and $4$ dense pairs. With the increased number of dense pairs from $1$ to $4$, the proposed method consistently increases performance as more semantically similar target pairs are extracted. Nevertheless, increasing the number of dense pairs from $4$ to $8$ will not continually improve the results since $4$ dense pairs might be enough to extract the learned dense representations using ViT-S/16. Furthermore, replacing $3 \times 3$ neighbors with all patches significantly deteriorates the performance of all metrics. These results indicate the importance of selecting semantically meaningful neighboring patches for capturing local semantics.

**Types of Local Aggregation Heads.** Local aggregation heads affect the ability of the proposed method to aggregate dense targets and context with local semantics. To explore such effects more comprehensively, we varied the type of Local Aggregation Heads from cross-attention and self-attention operators asymmetrically. We report the comparison results on the DAVIS benchmark with ViT-S/16 in Table 6. When both the context and target head use the cross-attention operators, we achieve the best performance in terms of all metrics. Replacing cross-attention operators with self-attention operators significantly worsens the results in terms of all metrics. These results indicate the difficulty in the asymmetric use of cross-attention aggregation heads, as the target aggregation head is updated using an exponential moving average of the context head weights, and it cannot be solely trained on its architecture.

## 5 CONCLUSION

We introduce a novel masked modeling objective tailored for the joint-embedding predictive architecture to learn better dense representations from unlabeled images. To tackle this, we aim to produce semantically meaningful target representations in a self-supervised manner by leveraging the prior that neighboring patches often contain similar semantics. To be specific, we first search semantically similar patches for a masked patch within its neighborhood by computing their similarities on the representation space. Then we generate the aggregated representations from the selected neighboring patches to serve as a masked modeling objective via a lightweight cross-attention head. Finally, the proposed objective would accelerate learned representations of semantically similar patches being closer, which can be advantageous in understanding local semantics within images. Through extensive experiments, we have demonstrated our models are not only effective in dense prediction types of downstream tasks but also show strong performance in image classification tasks. We believe that our work would not only highlight the effectiveness of considering a dense target for masked modeling on the embedding space but also provide a comprehensive understanding of local semantics within images through self-supervised pre-training.

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

## APPENDIX

In this appendix, we provide the following materials:

- Additional experiments on fine-tuning and segmentation & detection comparison with DINO (Caron et al., 2021) & MAE (He et al., 2021), and segmentation comparison with DINOv2 (Oquab et al., 2023) in Section A,

- Additional analyses on the effectiveness of cross-attention layers in local aggregation head, and computational comparison with I-JEPA (Assran et al., 2023) in Section B,

- Additional visualizations on the cosine similarity maps of query patches and learned attention maps in Section C.

## A    ADDITIONAL EXPERIMENTS

In order to further demonstrate the effectiveness of the proposed Dense-JEPA in learning local semantics during self-supervised pre-training, we conduct experiments on fine-tuning comparison with DINO (Caron et al., 2021) & MAE (He et al., 2021) on ImageNet-1K Deng et al. (2009), and segmentation & detection comparison on ADE20K (Zhou et al., 2017; 2018) & COCO (Lin et al., 2014) and segmentation comparison on ADE20K with DINOv2 (Oquab et al., 2023).

**Fine-tuning Comparison with DINO and MAE.** For a comprehensive comparison with DINO and MAE, we follow previous work (Caron et al., 2021; He et al., 2021) and fine-tune pre-trained ViT-B/16 & ViT-L/16 on ImageNet-1K. Table A.1 reports the comparison results with prior approaches using DINO and MAE pre-trained weights. As can be seen, our Dense-JEPA achieves the best performance in terms of all metrics for two different pre-trained models. These significant improvements demonstrate the superiority of our framework in learning the global semantics of images from self-supervised pre-training.

Table A.1: **Fine-tuning ImageNet-1K image classification.** We perform fine-tuning pre-trained ViT-B/16 for image classification on ImageNet-1K benchmark. We report the top-1 accuracy to evaluate the quality of fine-tuned representations. The best results are indicated in **bold**.

| Method | Pre-train data | Backbone | Top-1 Accuracy |
|---|---|---|---|
| DINO (Caron et al., 2021) | ImageNet-1K | ViT-B/16 | 82.8 |
| MAE (He et al., 2021) | ImageNet-1K | ViT-B/16 | 83.6 |
| Dense-JEPA (ours) | ImageNet-1K | ViT-B/16 | **84.6** |
| MAE (He et al., 2021) | ImageNet-1K | ViT-L/16 | 85.9 |
| Dense-JEPA (ours) | ImageNet-1K | ViT-L/16 | **86.6** |

**Segmentation & Detection Comparison with DINO and MAE.** For ADE20K segmentation & COCO detection, we follow the same setting in the previous work (Caron et al., 2021; He et al., 2021), and compare the proposed Dense-JEPA using ImageNet-1K pre-trained ViT-B/16 models in Table A.2. Compared to previous methods, we achieve the best results regarding all various metrics including mIoU, $AP^{box}$, and $AP^{mask}$. These results validate the effectiveness of our approach in learning local semantics during self-supervised pre-training.

Table A.2: **ADE20K semantic segmentation, COCO object detection, and instance segmentation.** We fine-tuned pre-trained ViT-B/16 models to perform ADE20K semantic segmentation and COCO object detection and instance segmentation. The mIoU, $AP^{box}$, and $AP^{mask}$ metrics denote the results of ADE20K segmentation, COCO detection, and COCO segmentation, respectively. The best results are indicated in **bold**.

| Method | Pre-train data | Backbone | mIoU | $AP^{box}$ | $AP^{mask}$ |
|---|---|---|---|---|---|
| DINO (Caron et al., 2021) | ImageNet-1K | ViT-B/16 | 46.8 | 50.1 | 43.4 |
| MAE (He et al., 2021) | ImageNet-1K | ViT-B/16 | 48.1 | 50.3 | 44.9 |
| Dense-JEPA (ours) | ImageNet-1K | ViT-B/16 | **49.0** | **50.9** | **45.6** |

**Segmentation Comparison with DINOv2.** For a more comprehensive comparison with DI-NOv2 (Oquab et al., 2023), the recent strong self-supervised baseline trained on a large-scale dataset, we compare our ImageNet pre-trained ViT-B/16 models with Figure 6 in the original DINOv2 paper using ImageNet pre-trained ViT-L/16 models on ADE20K semantic segmentation. The comparison results are reported in Table A.3. We can observe that our Dense-JEPA with ViT-B/16 backbone achieves better results than DINOv2 using ViT-L/16 using different pre-training resolutions. However, it should be noted that DINOv2 (Oquab et al., 2023) performed linear evaluation settings on the ADE20K dataset by freezing representations at different resolutions. Although the comparison is not totally fair, these results further demonstrate the competitive fine-tuning performance on segmentation by learning local semantics during self-supervised pre-training. We believe that our new framework can be scaled up to more pre-training data to achieve better downstream performances.

Table A.3: **ADE20K semantic segmentation.** We fine-tuned pre-trained ViT-B/16 models to perform ADE20K semantic segmentation. Note that DINOv2 (Oquab et al., 2023) performed linear evaluation settings on the ADE20K benchmark by freezing representations at different resolutions. The best results are indicated in **bold**.

| Method | Pre-train data | Backbone | Pre-train Resolution | mIoU |
|---|---|---|---|---|
| DINOv2 (Oquab et al., 2023) | ImageNet-1K | ViT-L/16 | 224 | 43.0 |
| | ImageNet-1K | ViT-L/16 | 416 | 46.2 |
| Dense-JEPA (ours) | ImageNet-1K | ViT-B/16 | 224 | **49.0** |

## B    ADDITIONAL ANALYSIS

In this section, we performed ablation studies to demonstrate the advantage of using cross-attention layers against pooling operators in the proposed context and target aggregation heads in Local Aggregation Target modules, and computational costs for training. Our ablation experiments are based on ImageNet-1k pre-trained ViT-S/16 models.

**Ablation on Cross-attention vs Pooling in Local Aggregation Head.** To validate the effectiveness of using cross-attention layers, we ablated the layer using average-pooling and max-pooling operators. The comparison results are reported in Table B.1. As can be observed, replacing the cross-attention layer with average-pooling deteriorates the results by $1.9@(\mathcal{J}\&\mathcal{F})_m$, $1.4@\mathcal{J}_m$, and $2.4@\mathcal{F}_m$. This might be because average pooling leads to collapsing and losing discrepancy across patch tokens. Our key is each query patch has distinct neighborhood patches, and objectively derived from the distinct neighborhoods will ensure their distinct target objectives. Meanwhile, using the max-pooling operator highly decreases the results in terms of all metrics. These results validate the importance of cross-attention layers in preventing losing discrepancy for distinct target representations during self-supervised pre-training.

Table B.1: **Ablation studies on local aggregation head.** We perform ablation studies on the cross-attention layers in the Local Aggregation Target module using ViT-B/16 pre-trained models. The best results are indicated in **bold**.

| Local Aggregation Head | $(\mathcal{J}\&\mathcal{F})_m$ | $\mathcal{J}_m$ | $\mathcal{F}_m$ |
|---|---|---|---|
| Average-pooling | 55.2 | 54.3 | 56.1 |
| Max-pooling | 55.6 | 54.6 | 56.6 |
| Cross-attention | **57.1** | **55.7** | **58.5** |

**Computational Comparison with I-JEPA.** In order to comprehensively assess the efficiency of the proposed Dense-JEPA, we compared it with I-JEPA (Assran et al., 2023), the state-of-the-art image-based joint-embedding predictive architecture, on total pre-training epochs, max memory per GPU, and running time per step in Table B.2. We can observe that our Dense-JEPA achieves comparable computation costs in terms of all metrics, especially on total pre-training epochs and max memory usage. More importantly, we achieve much better downstream performance regarding segmentation & detection in Table 1 & 2, image classification in Table 3 & A.1, and other low-level tasks in Table 4. These computational analyses further demonstrate the efficiency of our novel framework.

Table B.2: **Analysis on computational costs.** We perform computational analyses on pre-trained ViT-B/16 models for comparison with I-JEPA. The best results are indicated in **bold**.

| Method | Pre-train Epochs | Max Memory per GPU (GB) | Running Time per Step (ms) |
|---|---|---|---|
| I-JEPA (Assran et al., 2023) | 600 | 21.9 | 606.4 |
| Dense-JEPA (ours) | 600 | 21.9 | 608.2 |

## C  ADDITIONAL VISUALIZATIONS

In order to qualitatively demonstrate the effectiveness of the proposed Dense-JEPA in learning local semantics during pre-training, we provide learned attention maps from the target encoder using pre-trained I-JEPA and our method, and attention maps from the cross-attention layer in the proposed Local Aggregation Target (LAT) modules in Figure C.1, C.2, C.3, and C.4. Furthermore, we visualize the top-10% of highly correlated patches in more examples by thresholding the cosine-similarity maps of query patches in the last layer in Figure C.5, C.6, C.7, and C.8.

**Learned Attention Maps.** Learning discriminative attention maps is one of the key aspects of capturing local semantics for downstream tasks of dense prediction type, such as segmentation and detection. To better evaluate the quality of learned attention maps, we visualize the averaged maps from different heads in the last attention layer by using a pre-trained ViT-B/16 target encoder. For a more comprehensive comparison, we also add the attention maps from I-JEPA (Assran et al., 2023) target encoder and the cross-attention layer in our Local Aggregation Target (LAT) module. The qualitative visualization results are shown in Figure C.1, C.2, C.3, and C.4. Note that columns for each image sample represent the original image, attention maps from the target encoder in I-JEPA, attention maps from the target encoder in our Dense-JEPA, and attention maps from the local aggregation target module in our Dense-JEPA. As can be seen, both attention maps from the target encoder and the local aggregation target module in our Dense-JEPA are discriminative and focus on the object semantics in the image. However, the attention maps from the target encoder in I-JEPA activate both the foreground and background and can not effectively discriminate the object semantics, because they did not incorporate local semantics into target representations as our Dense-JEPA did. Meanwhile, the attention maps of the local aggregation target module in our Dense-JEPA have more focus on the details of foreground objects than that from the target encoder, indicating the effectiveness of our local aggregation target module in generating target representations with local semantics. These high-quality visualization results further demonstrate the superiority of our new framework in learning meaningful representations with local semantics during self-supervised training, compared to I-JEPA, the state-of-the-art image-based joint-embedding predictive architecture.

**Learned Cosine Similarity Maps.** To further validate the effectiveness of our method in learning dense representations, we visualize the top-10% of highly correlated patches by thresholding the cosine similarity maps of query patches in the last layer using the pre-trained ViT-B/16 target encoder. Figure C.5, C.6, C.7, and C.8 showcase the qualitative visualization results, where rows for each sample denote the location of the given query patch and the top-10% patches. We can observe that the patches extracted by our Dense-JEPA are centralized and specifically focus on the location of the given query patch. For example, given the first query patch on the building in the car example shown in Figure C.7, the top-10% patches focus on the building. When the query patch is given on the location of the car, the top-10% patches also change to focus on the car. Another example can also be seen in the dynamic changes with respect to the location of query patches on the head, arm, elbow, and microphone in the first sample shown in Figure C.8. Interestingly, when the query patch is given on the body part from one of two birds in the second example shown in Figure C.7, the top-10% patches can highlight the location of body parts from both birds, which might be due to the similar local semantics in both body locations. These visualization results demonstrate that our Dense-JEPA performs effectively by encouraging the model to learn dense representations.

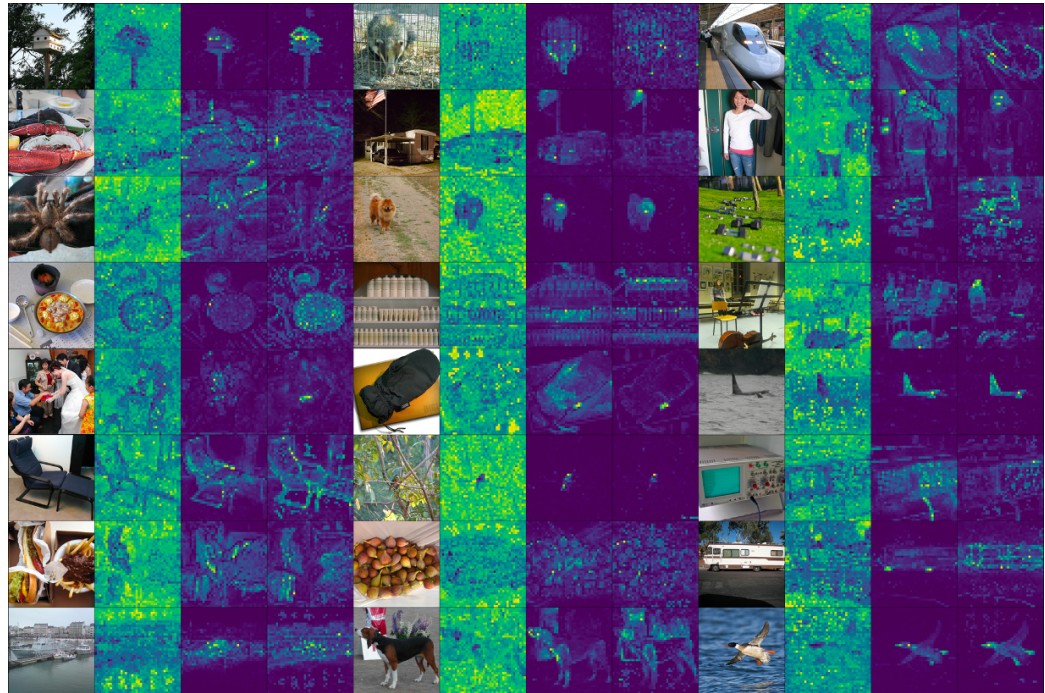

Figure C.1: Qualtitative visualization of learned attention maps using ViT-B/16 model. Columns for each sample denote the original image, attention maps from target encoder in I-JEPA (Assran et al., 2023), attention maps from target encoder in our Dense-JEPA, and attention maps from the local aggregation head in our Dense-JEPA. Our Dense-JEPA achieves much better attention maps.

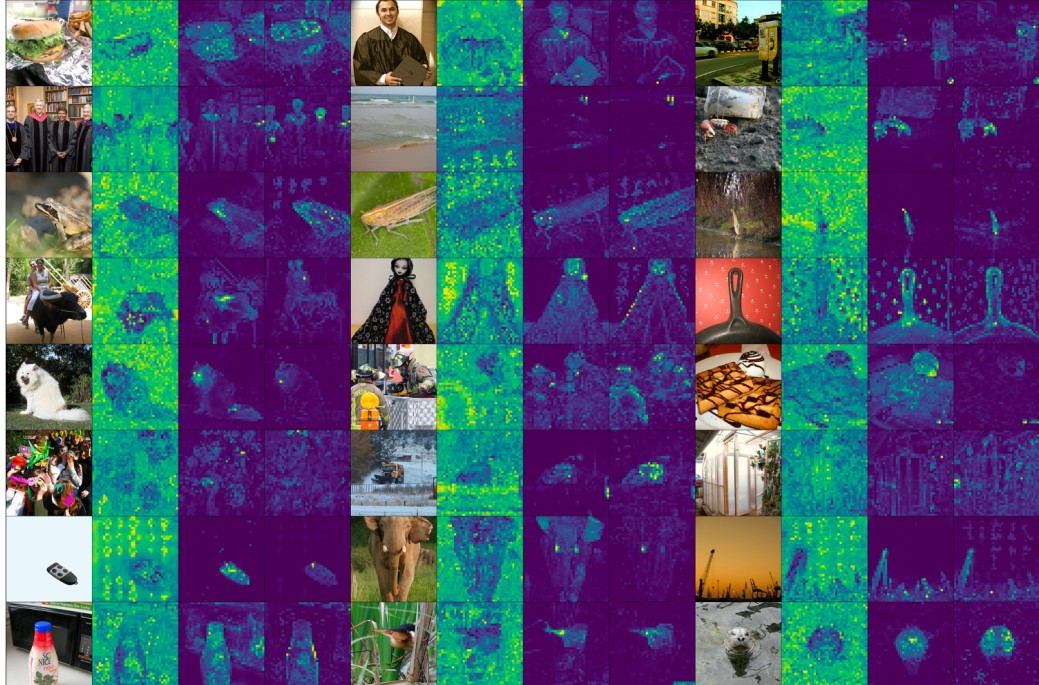

Figure C.2: Qualtitative visualization of learned attention maps using ViT-B/16 model. Columns for each sample denote the original image, attention maps from target encoder in I-JEPA (Assran et al., 2023), attention maps from target encoder in our Dense-JEPA, and attention maps from the local aggregation head in our Dense-JEPA. Our Dense-JEPA achieves much better attention maps.

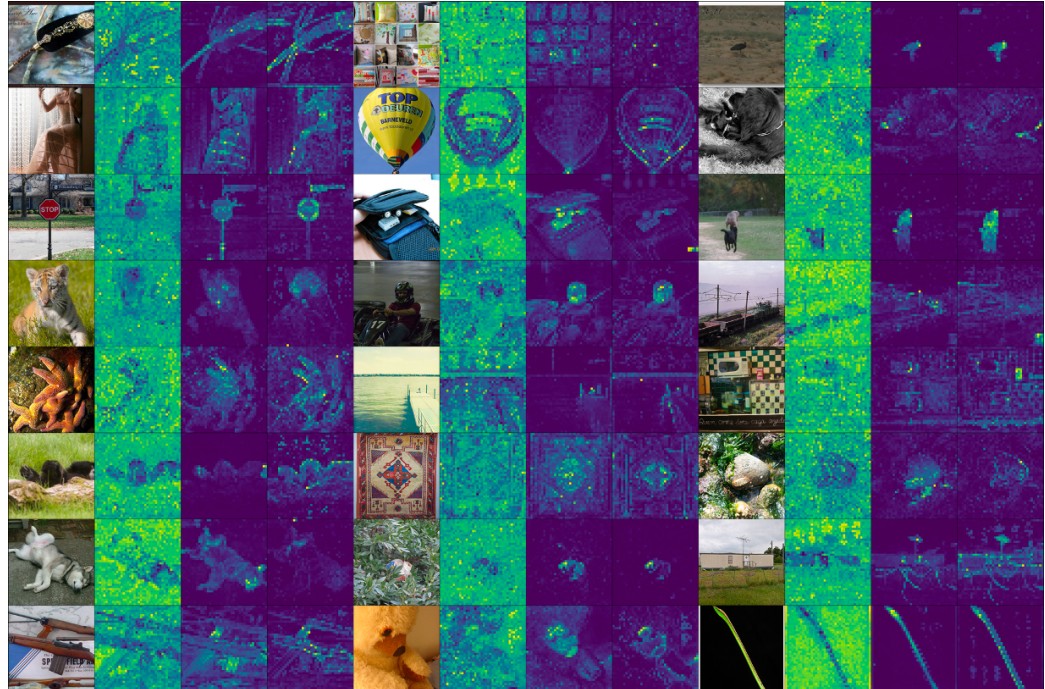

Figure C.3: Qualtitative visualization of learned attention maps using ViT-B/16 model. Columns for each sample denote the original image, attention maps from target encoder in I-JEPA (Assran et al., 2023), attention maps from target encoder in our Dense-JEPA, and attention maps from the local aggregation head in our Dense-JEPA. Our Dense-JEPA achieves much better attention maps.

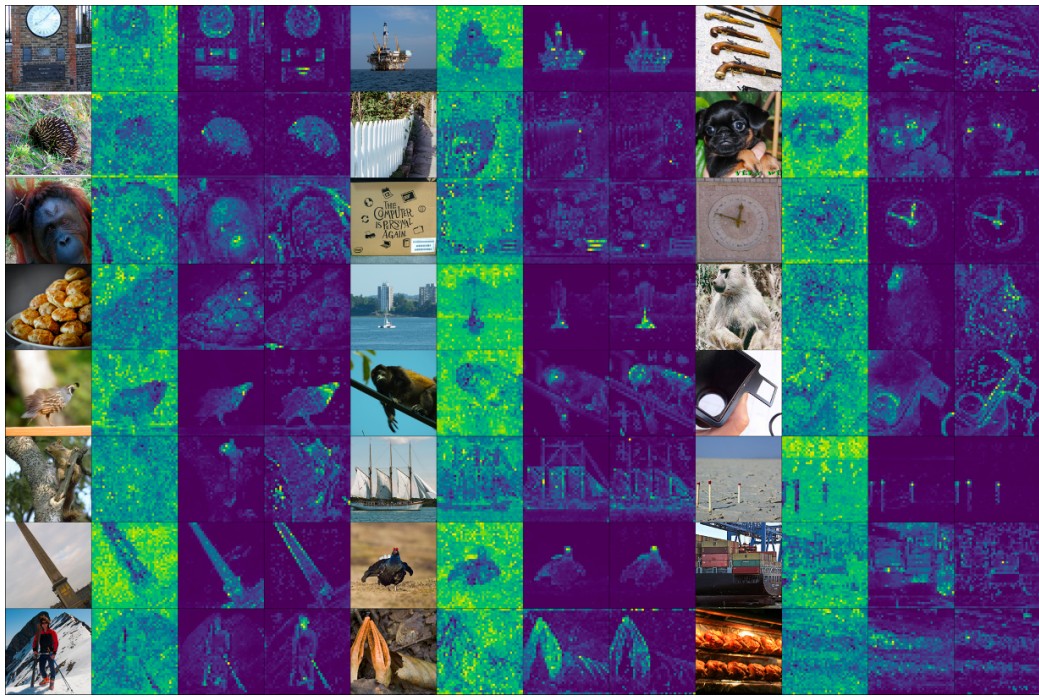

Figure C.4: Qualtitative visualization of learned attention maps using ViT-B/16 model. Columns for each sample denote the original image, attention maps from target encoder in I-JEPA (Assran et al., 2023), attention maps from target encoder in our Dense-JEPA, and attention maps from the local aggregation head in our Dense-JEPA. Our Dense-JEPA achieves much better attention maps.

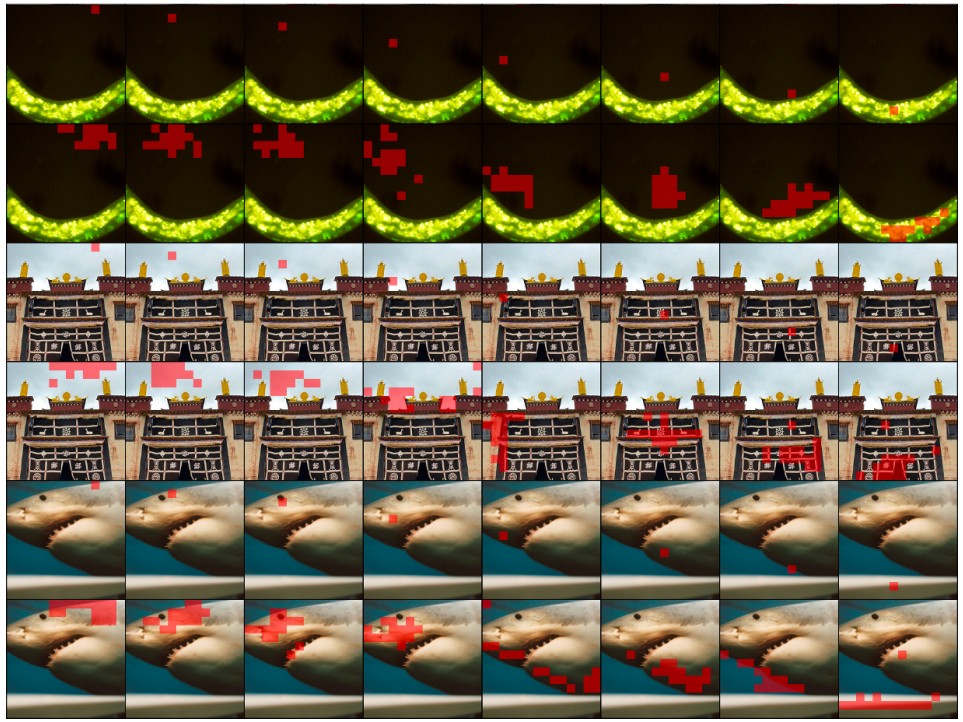

Figure C.5: Qualtitative visualization of learned cosine similarity maps given query patches using ViT-B/16. Rows for each sample denote the location of the given query patch and the top-10% patches. Our Dense-JEPA performs effectively by encouraging the model to learn local semantics.

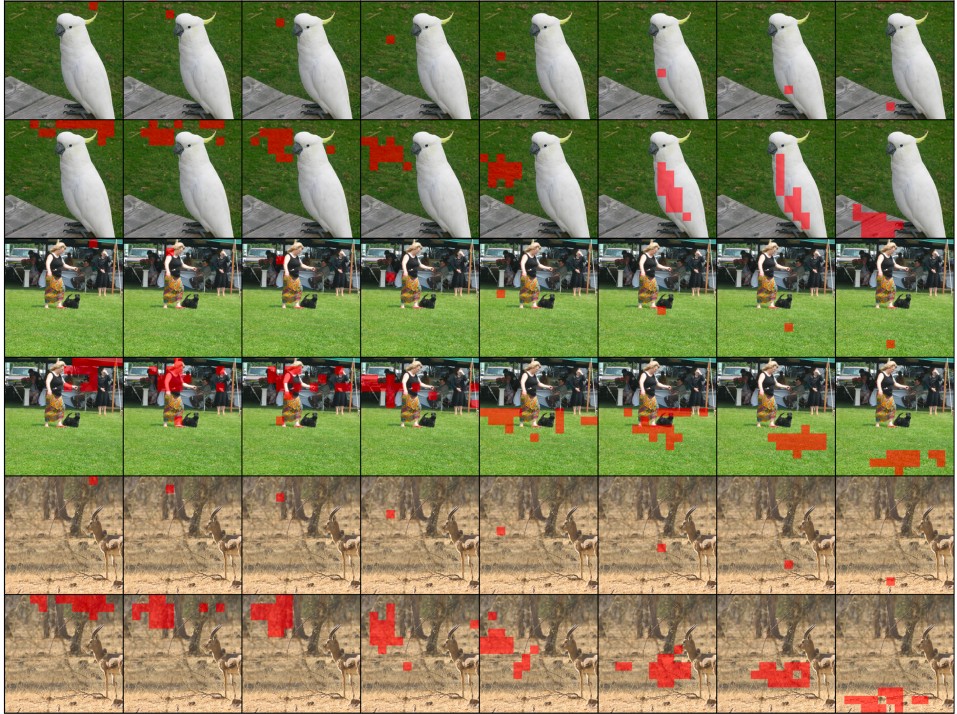

Figure C.6: Qualtitative visualization of learned cosine similarity maps given query patches using ViT-B/16. Rows for each sample denote the location of the given query patch and the top-10% patches. Our Dense-JEPA performs effectively by encouraging the model to learn local semantics.

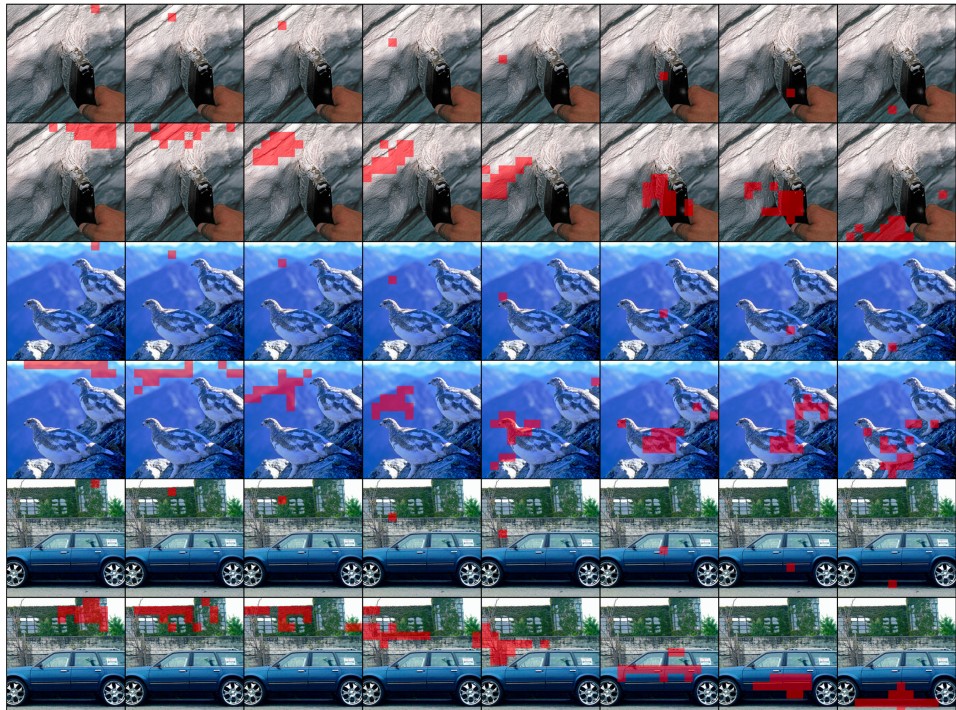

Figure C.7: Qualtitative visualization of learned cosine similarity maps given query patches using ViT-B/16. Rows for each sample denote the location of the given query patch and the top-10% patches. Our Dense-JEPA performs effectively by encouraging the model to learn local semantics.

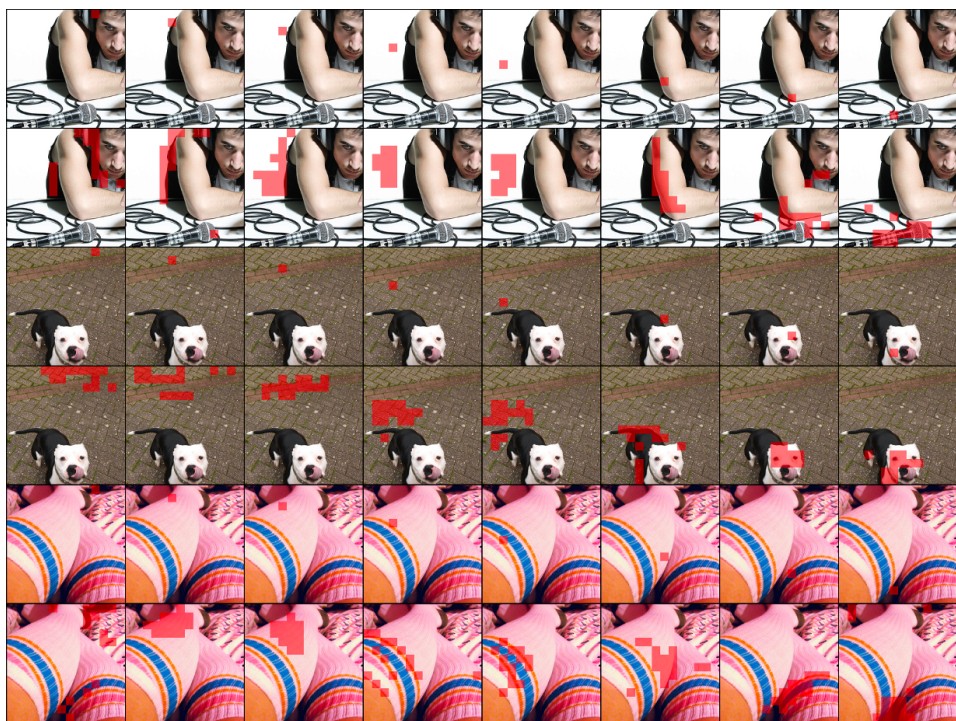

Figure C.8: Qualtitative visualization of learned cosine similarity maps given query patches using ViT-B/16. Rows for each sample denote the location of the given query patch and the top-10% patches. Our Dense-JEPA performs effectively by encouraging the model to learn local semantics.

