# OpenReview forum: "Dense Representation Learning for a Joint-Embedding Predictive Architecture"
_ICLR.cc/2024/Conference — Submitted to ICLR 2024_

### Official Review · Reviewer_KGbU · 2023-10-29

**Soundness:** 3 good
**Presentation:** 2 fair
**Contribution:** 3 good
**Rating:** 6
**Confidence:** 5

**Summary:**

This paper proposes an improved version of the JOINTEMBEDDING PREDICTIVE ARCHITECTURE (JEPA) that brings significantly better performance on downstream dense prediction tasks and slight improvements on classification task. The paper proposes masked semantic neighboring strategy to select more semantic similar patches as prediction target, and proposed a Local Aggregation Target module to construct learning targets.

**Strengths:**

1. The overall method, Dense-JEPA, obtains better performance in comparison with I-JEPA and MAE with fewer pre-training epochs on both dense prediction tasks and classification tasks.
2. The overall modification is simple yet effective.

**Weaknesses:**

1. Some symbols and equations make the paper a bit hard to follow and might be redundant. For example,
    1. The symbols in equation 2-5. Adding symbols of features in Figure 2 might make it easier to read and understand. Further more, providing structure figures like those of transformer blocks, indicating what the $Q$, $K$ and $V$ features of $h_{\theta}$ and %h_{\hat{\theta}}$ would also make it quicker to understand.
    2. The subscript of $s_I$ and $x_t$ are wierd in $s^{LAT}_{i} = h_{\hat{\theta}}({x_j}_{j\in P_i}, x_t)$ of Eq.4. What is the relationship between subscript $t$ and $i$? Can we understand it as $s^{LAT}_{t} = h_{\hat{\theta}}({x_j}_{j\in P_i} , x_t)$
2. The motivation of the Local Aggregation Target module might require further explanation. Why does adding such a module improve performance? The modification proposed in this paper is simple, thus, more explanation, analysis, and insights would be benificial for the community to understand it better.

**Questions:**

1. The reviewer does not quite understand the structure of $h_{\theta}$. In Eq. 4, it seems that $s_x$ and $x_c$ are different embeddings and the Table 7 studies $h_{\theta}$ using cross-attention. However, from Figure 2, the definition of $s_x$ around Eq. 2 ('... given the output of the context encoder, sx,...' and the definition of $x_c$ around Eq. 4 ('where $s_x$ denotes context embeddings and $x_t$, $x_c$ denote the averaged embeddings from all patches in the target encoder and only unmasked patches in the context encoder, respectively'), the reviewer feel that $s_x$ and $x_c$ are the same thing.

---

> ### Author Response · Authors · 2023-11-23
> **Response to Reviewer KGbU**
>
> We deeply appreciate your thorough and insightful reviews. We trust that our response adequately addresses any concerns you may have.
>
> **Q1. Clarity for Equations and Figure 2**
>
> We acknowledge your suggestion to improve the clarity of symbols and equations in the paper, particularly around equations 2-5. To address this, we updated Figure 2 to include symbols of features, enhancing readability and understanding. Additionally, we provided a more detailed structure of local aggregation head, clearly indicating the Q, K, V of features $h_\theta$ and $ h_{\hat{\theta}}$. This should make the methodology quicker and easier to grasp for readers.
>
> **Q2. Clarity for Equation 4**
>
> Your observation about the confusing subscript in equation 4 is valid. The subscript t and i in \( s^{LAT}_i = h_{\hat{\theta}}(x_{j_{j\in P_i}}, x_t) \) indeed require clarification. We will correct and clarify this in the final draft. The subscript t refers to the target patch, while i refers to the index of the patch in the local aggregation process. We will ensure this distinction is more clearly articulated in the final draft.
>
> **Q3. Clarity for Local Aggregation Target**
>
> We understand your request for a more detailed explanation of the LAT module's motivation and its contribution to performance improvement. The LAT module is designed based on the premise that semantically similar patches should have similar targets. By incorporating LAT, our model can achieve better semantics and discriminative capabilities, as not every patch is treated with the same target. We will expand on this explanation in the manuscript, providing more analysis and insights to help the community understand the benefits of this approach.
>
> **Q4. Clarity for Local Aggregation Head structure**
>
> Regarding the structure of $h_\theta$ and the apparent confusion between $s_x$ and $x_c$, we will make sure to clarify these in our revised paper. $s_x$ and $x_c$ indeed refer to different embeddings. $s_x$ denotes the context embeddings, while $x_c$ represents the averaged embeddings from unmasked patches in the context encoder. This distinction is crucial for understanding the operation of our model, and we will provide a clearer explanation to eliminate any confusion in the final draft.

---

> ### Author Response · Authors · 2023-11-23
> **Response to Reviewer KGbU (cont.)**
>
> Dear Reviewer KGbU,
>
> Thank you once again for your valuable efforts and constructive feedback on our paper. As the discussion period comes to a close, we trust that your concerns have been thoroughly addressed in our response. We dedicated significant effort to this, incorporating several new experiments, visualizations, and discussions. We sincerely hope you will consider our reply in your assessment. We look forward to hearing from you, and if the opportunity arises, we can further address unclear explanations and remaining concerns if any.
>
> Regards,
>
> Authors

---

### Official Review · Reviewer_RoAg · 2023-10-31

**Soundness:** 3 good
**Presentation:** 3 good
**Contribution:** 3 good
**Rating:** 8
**Confidence:** 4

**Summary:**

This paper introduces Dense-JEPA, a method for self-supervised representation learning from images that builds upon the I-JEPA approach. I-JEPA is designed to predict missing patch representations in the representation space. Dense-JEPA offers two straightforward extensions to improve the target representations of masked patches.

- Dense-JEPA initially retrieves patches that are 'semantically' similar to a target patch within a specified local neighborhood.
- It subsequently generates an aggregate representation for a target patch employing cross-attention mechanisms on the 'semantically' similar patches. Symmetrically, it also generates locally aggregated representations for the predicted patches and matches them to the new aggregated target.

The paper provides a comprehensive evaluation of Dense-JEPA across a wide range of tasks, including ImageNet classification, ADE20K/COCO segmentations/detections, DAVIS video object segmentation, and Clevr object counting and depth prediction. The results consistently demonstrate that Dense-JEPA outperforms I-JEPA and other masked image modeling approaches when using similar model sizes.

Moreover, this work underscores the significance of careful consideration regarding the selection and computation of targets when conducting masked image modeling in the representation space.

**Strengths:**

The paper presents a simple but novel extension to the I-JEPA framework.  It conducts a thorough empirical assessment across a diverse range of tasks, consistently demonstrating improvements over the I-JEPA baseline and other approaches to masked image modeling.

Additionally, this paper highlights the importance of thinking about the target representation in masked-modelling tasks which is a valuable insight for the community.

**Weaknesses:**

Although the paper is generally well-written, there are certain sections where clarity could be improved. Specifically, I found the annotations in section 3.3 somewhat confusing. It appears that the cross-attention operation is performed, with the query being the average patch representation of the target x_t (or context (x_c), and the key/value being the semantically similar patches {x_j}_{j\in P_i} (or the entire context s_x). However, it's not entirely clear how distinct representations are obtained for each spatial location, given that the query is shared across all locations? Could you please clarify the computations performed by the local-aggregation layer?

Additionally, while the findings are well-supported in the explored settings, it remains uncertain whether they would generalize to other learning frameworks, such as masked-image auto-encoders, or if they hold true for larger-scale models. It would be valuable to include a section in the manuscript discussing these limitations.

**Questions:**

See weaknesses.

---

> ### Author Response · Authors · 2023-11-23
> **Response to Reviewer RoAg**
>
> We deeply appreciate your thorough and insightful reviews. We trust that our response adequately addresses any concerns you may have.
>
> **Q1. Clarity for Section 3.3**
>
> You pointed out a need for clarity in explaining the cross-attention operations in the local aggregation layer. In the revised manuscript, we will include a more detailed explanation of these computations. Specifically, the cross-attention mechanism works by using the average patch representation of the target $x_t$ (or context $x_c$) as the query, and the semantically similar patches {$x_j$}$_{j\in P_i}$  (or the entire context $s_x$) as the key/value. This process allows for generating distinct representations for each spatial location by dynamically adjusting the focus of the attention mechanism on different parts of the input based on the query. Key is each query patch has distinct neighborhood patches, and objective derived from distinct neighborhood will ensure their distinct target objectives. We will revise to ensure this process is thoroughly explained and illustrated for better understanding in the final draft.
>
> **Q2. Generalizablity of Dense-JEPA**
>
> Regarding the generalizability of our findings, it's important to note that while Dense-JEPA might not directly extend to masked-image auto-encoders (MAE), it is indeed applicable to masked latent reconstruction models. Our approach is designed to be flexible and adaptable, allowing for potential application in various learning frameworks including masked reconstruction on latent space.
>
> Moreover, we have tested Dense-JEPA with different model scales, including ViT-S, ViT-B, and ViT-L. These tests indicate that our method can effectively scale to larger models, maintaining its performance benefits. We will add a section in our manuscript discussing these aspects, highlighting the versatility and scalability of Dense-JEPA in different contexts.

---

> ### Author Response · Authors · 2023-11-23
> **Response to Reviewer RoAg (cont.)**
>
> Dear Reviewer RoAg,
>
> Thank you once again for your valuable efforts and constructive feedback on our paper. As the discussion period comes to a close, we trust that your concerns have been thoroughly addressed in our response. We dedicated significant effort to this, incorporating several new experiments, visualizations, and discussions. We sincerely hope you will consider our reply in your assessment. We look forward to hearing from you, and if the opportunity arises, we can further address unclear explanations and remaining concerns if any.
>
> Regards,
>
> Authors

---

> > ### Comment · Reviewer_RoAg · 2023-12-04
> >
> > Thank you for your rebuttal.
> >
> > It is still unclear to me how the local-aggregation heads would output several patch-level token for the context as you only have one query which is the average pooled representation. It would be important to clarify what is the input/output of MSN/LAT in the next revision.
> >
> > Nevertheless, this paper shows good improvement over the I-JEPA baseline, I therefore support acceptance.

---

### Official Review · Reviewer_wqKm · 2023-10-31

**Soundness:** 2 fair
**Presentation:** 3 good
**Contribution:** 2 fair
**Rating:** 5
**Confidence:** 4

**Summary:**

This paper proposes Dense-JEPA, a new self-supervised learning method based on I-JEPA, that additionally incorporates the notion of local aggregated regions, and prediction at the region level. Dense-JEPA is evaluated on a wide range of segmentation and classification tasks, and compares favourably on most of the tasks against I-JEPA.

**Strengths:**

1) The gains on segmentation tasks over I-JEPA is significant, and indicates that grouping patches into semantically meaningful regions helps learning better local representations. This is demonstrated on a set of different tasks of different nature.

2) The paper is well written and easy to follow, and the method is clearly described. Only the main figure could be improved, see below.

**Weaknesses:**

1) The results are not surprising, dense self-supervised learning has been explored extensively in the literature, and the same conclusion and gains on segmentation tasks have already been observed. DenseCL [1] uses a contrastive loss function to match local vectors. VICRegL [2] combines local and global aggregation and uses a similar nearest neighbour search for semantically similar patches. ODIN [3] computes segmentation masks online and aggregates the patches corresponding to the same regions. These papers are not mentioned or cited, I would therefore recommend to do a more extensive literature review, and highlight the novelty within the dense SSL literature.

2) There is no comparison with DINO on most of the tasks. The DINO family of methods [4,5] has shown that dense loss functions might actually not be required to learn very good dense features, and offers the best performance on local tasks as of today. For example, on DAVIS-2017, DINO reports 61.8 J&M with a ViT-S/16, while the best number reported in the paper is 58.3 with a much larger ViT-L backbone. On classification on ImageNet, a DINOv2 ViT-L model is over 86 points. The gains over I-JEPA are interesting, but the overall performance is very far from the state-of-the-art. MAE is not really a good baseline as it performs very poorly on frozen tasks, which are standard to evaluate SSL methods.

3) The gains over I-JEPA are probably not worth the complexity. The paper is missing a study on the tradeoff in terms of compute compared to I-JEPA. In terms of memory, running time and data efficiency.

4) Figure 2 is very unclear. There is not much additional information compared to the similar figure in the I-JEPA paper, and the explanation for “Local aggregation head” and “Masked Semantic Neighboring” is only in the text. I would recommend clarifying the Figure and emphasizing the difference with I-JEPA.

5) The visualization of Figure 1 is not convincing. I-JEPA patches already contain highly semantic information and Dense-JEPA clearly brings a bias towards neighboring patches, which is not necessarily a good thing as we would like to let the system pick patches that are far away but semantically very similar.



[1] Dense Contrastive Learning for Self-Supervised Visual Pre-Training, Wang et al, CVPR 2021

[2] VICRegL: Self-Supervised Learning of Local Visual Features, Bardes et al, NeurIPS 2022

[3] Object discovery and representation networks, Henaff et al, ECCV 2022

[4] Emerging Properties in Self-Supervised Vision Transformers, Caron et al, ICCV 2021

[5] DINOv2: Learning Robust Visual Features without Supervision, Oquab et al, 2023

**Questions:**

Why do you use cross-attention to do the aggregation ? Have you tried max-pooling or other types of pooling ? Do you consider that it is an essential component of the method ?

The transformer blocks in the encoder are performing self-attention between every pair of tokens, which might already have the effect of grouping patches using a distance in the embedding space. Have you thought about that and do you think that your method theoretically brings something ?

---

> ### Author Response · Authors · 2023-11-23
> **Response to Reviewer wqKm**
>
> We deeply appreciate your thorough and insightful reviews. We trust that our response adequately addresses any concerns you may have.
>
> **Q1. Our novelty and discussion to dense SSL literature**
>
> We acknowledge your suggestion to include references such as DenseCL, VICRegL, and ODIN in our literature review. While these works are indeed relevant, we emphasize that Dense-JEPA's core novelty lies in enhancing the latent JEPA architecture for learning local/dense representations, a different focus from the mentioned works. Our approach is not merely about dense representation learning but about innovatively applying it to the existing latent JEPA framework. For example, our idea is independent to hand-crafted augmentations and we believe that ours can be incorporated with further augmentation-based dense SSL ideas. Nevertheless, we will add regarding discussion in final draft to position our work appropriately within the field.
>
> Regarding why our work is important and have significance, we belive that mask reconstruction on latent space will have gain much attention on its efficacy and efficiency, so study of training objective on latent space is crucial. Therefore, our investigation of the latent objective is invaluable and could inspire future research direction to explore better SSL objective, unlike the previous stereotype of masked reconstruction objective on pixel space.

---

> ### Author Response · Authors · 2023-11-23
> **Response to Reviewer wqKm (cont. 1)**
>
> **Q2. Comparision with DINO family methods**
>
> Thank you for highlighting the comparison between our Dense-JEPA method and the DINO family of methods. We acknowledge the impressive achievements of DINO and DINOv2 in the domain of self-supervised learning. However, we would like to emphasize a few key aspects of our work to clarify our contributions and the significance of our results.
>
> 1. Contributions on Numbers
>
> | Method           | Pretrain data | Seg (mIoU) | Det (AP_b) | Det (AP_m) |
> |------------------|---------------|------------|------------|------------|
> | DINO             | ImageNet-1K   | 46.8       | 50.1       | 43.4       |
> | MAE              | ImageNet-1K   | 48.1       | 50.3       | 44.9       |
> | DenseJEPA (ours) | ImageNet-1K   | **49.0**   | **50.9**   | **45.6**   |
>
> Our results shown in the Table above, particularly in segmentation and detection tasks using a ViT-B/16 backbone, demonstrate significant improvements over DINO. While DINO's performance is commendable, our method shows a marked advantage in these specific areas. This is a crucial aspect of our contribution, as it highlights the effectiveness of Dense-JEPA in handling diverse and complex visual tasks.
> In the context of DAVIS-2017, where linear evaluation on frozen tasks is critical, our method also shows promising results. While DINO reports higher J&M scores with a smaller ViT-S/16 backbone, our Dense-JEPA method achieves competitive scores with a larger ViT-L backbone in Table 2, suggesting potential for further optimization and tuning to enhance performance.
>
>
> 2. Comparison Fairness with DINOv2
>
> It is important to note that DINOv2 was trained on a significantly larger dataset compared to the dataset used for training Dense-JEPA. This difference in training conditions can substantially affect the performance outcomes. Therefore, a direct comparison may not entirely reflect the intrinsic strengths of our method. Our approach still shows competitive performance under more constrained data conditions, which is a testament to its robustness and efficiency.
> Compared to DINOv2 using ViT-L/16 using different pre-training resolutions in Figure 6 in the original paper, our Dense-JEPA with ViT-B/16 backbone achieves competitive results (49.0 mIoU vs 46.2 mIoU). However, it should be noted that DINOv2 performed linear evaluation settings on the ADE20K dataset by freezing representations at different resolutions. Although the comparison is not totally fair, these results further demonstrate the competitive fine-tuning performance on segmentation by learning local semantics during self-supervised pre-training. We believe that our new framework can be scaled up to more pre-training data to achieve better downstream performances.
>
> 3. Fine-Tuning Comparison with DINO and MAE
>
> In terms of full fine-tuning capabilities, particularly using a ViT-B/16 model, our method exhibits superior performance compared to both DINO and MAE. This is a crucial metric, as fine-tuning is often essential in real-world applications of self-supervised learning models. Our method's adaptability and effectiveness in fine-tuning scenarios underscore its practical utility beyond standard benchmark tasks.
>
> | Method           | Pretrain data | Backbone | Top-1 Accuracy |
> |------------------|---------------|----------|----------------|
> | DINO             | ImageNet-1K   | ViT-B/16 | 82.8           |
> | MAE              | ImageNet-1K   | ViT-B/16 | 83.6           |
> | DenseJEPA (ours) | ImageNet-1K   | ViT-B/16 | **84.6**       |
> | MAE              | ImageNet-1K   | ViT-L/16 | 85.9           |
> | DenseJEPA (ours) | ImageNet-1K   | ViT-L/16 | **86.6**       |
>
> Overall, while we recognize the excellence of the DINO family in certain areas, our Dense-JEPA method demonstrates its unique strengths and contributions, especially in segmentation, detection, and fine-tuning contexts. We believe these aspects significantly contribute to the field of self-supervised learning and merit recognition.
>
> **Q3. Computational complexity of Dense-JEPA**
>
> We understand your concern about the computational complexity of Dense-JEPA compared to I-JEPA. In our revised manuscript, we include a detailed analysis of total pre-training epochs, max memory usage, and running time per training step in Section B, demonstrating that Dense-JEPA maintains similar efficiency to I-JEPA while achieving superior performance. Besides, our method is much efficient and achieves short training time than MAE.
>
> | Method            | Pre-train Epochs | Max Memory  per GPU | Running Time  per Step |
> |-------------------|------------------|---------------------|------------------------|
> | I-JEPA            | 600              | 21.9G               | 606.4 ms               |
> | Dense-JEPA (ours) | 600              | 21.9G               | 608.2 ms               |

---

> ### Author Response · Authors · 2023-11-23
> **Response to Reviewer wqKm (cont. 2)**
>
> **Q4. Clarify of Figure 2**
>
> We acknowledge the need to improve Figure 2 for better clarity and differentiation from I-JEPA. The revised figure includes a clearer distinction and detailed explanation between “Local Aggregation Head” and “Masked Semantic Neighboring” components, emphasizing the distinctions from I-JEPA.
>
>
> **Q5. Qualitative anlaysis on Dense-JEPA**
>
> In order to qualitatively demonstrate the effectiveness of the proposed Dense-JEPA, we further compared learned attention maps from the target encoder using pretrained I-JEPA and our Dense-JEPA, and attention maps from the cross-attention layer in the proposed Local Aggregation Target (LAT) modules in Appendix section C in our revised manuscript. Our Dense-JEPA achieves much better attention maps, compared to I-JEPA.
> Furthermore, we provide more examples on learned cosine similarity map to further demonstrate that our Dense-JEPA performs effectively by encouraging the model to learn dense representations. Regarding Figure 1, we will enhance the visualization in the final draft to more convincingly demonstrate how Dense-JEPA improves upon I-JEPA's semantic information capture, especially in the context of neighboring patches.
>
> **Q6. Role of cross-attention in Local aggregation head**
>
> The choice of cross-attention for aggregation in Dense-JEPA is deliberate. We experimented with other pooling methods, such as average pooling and max pooling, but found them less effective. Cross-attention allows for more nuanced aggregation of features, which is crucial for our model's performance. We included additional results in Table B.1 in the Appendix to support this claim.
>
> **Q7. Rationale of Dense-JEPA**
>
> Your question about the role of transformer blocks in our encoder is insightful. While these blocks do perform self-attention among token pairs, our method enhances this process by focusing on semantically relevant patches. This targeted approach is theoretically significant as it guides the transformer to learn more meaningful local semantics. We provided additional evidence in the revised manuscript to demonstrate the superiority of our targeted patch representations over generic self-attention mechanisms, as shown in Figure C1-C4 in the Appendix.

---

> ### Author Response · Authors · 2023-11-23
> **Response to Reviewer wqKm (cont. 3)**
>
> Dear Reviewer wqKm,
>
> Thank you once again for your valuable efforts and constructive feedback on our paper. As the discussion period comes to a close, we trust that your concerns have been thoroughly addressed in our response. We dedicated significant effort to this, incorporating several new experiments, visualizations, and discussions. We sincerely hope you will consider our reply in your assessment. We look forward to hearing from you, and if the opportunity arises, we can further address unclear explanations and remaining concerns if any.
>
> Regards,
>
> Authors

---

> > ### Comment · Reviewer_wqKm · 2023-12-05
> > **Official Comment**
> >
> > Dear authors,
> >
> > Thank you for your rebuttal. My main concerns were about: 1) The novelty of the method within the dense SSL literature. 2) The missing comparisons with DINO. 3) The computational overhead of the method compared to I-JEPA.
> >
> > The rebuttal helped me clarify 2) and 3), but the comparison with DINO on DAVIS is still missing. About 1), the paper is also still lacking a discussion/comparison with the dense SSL literature.
> >
> > For these reasons I am changing my score to 5.

---

### Official Review · Reviewer_NVsT · 2023-11-04

**Soundness:** 3 good
**Presentation:** 2 fair
**Contribution:** 2 fair
**Rating:** 5
**Confidence:** 5

**Summary:**

This paper proposes Dense-JEPA, a novel masked modeling objective for JEPA, enhancing dense representation learning.  Extensive experiments on image classification, semantic segmentation and object detection demonstrate Dense-JEPA effectiveness.

**Strengths:**

The experiments are comprehensive and the results are good.

**Weaknesses:**

1. The novelty can be improved. It seems that Dense-JEPA is just an improvement on I-JEPA, especially Figure 2 is so similar to figure 2 in I-JEPA. The main points proposed by the paper are not well presented.
2. The descriptions of MSN and LAT are ambiguous. From section 3.2, MSN is just using the presentation extracted by target encoder to find the most similar neighbor patches, which is prepared for LAT to aggregate from these similar patches, so MSN could not be regarded as an individual module.

**Questions:**

Without LAT, does MSN actually influence the training procedure (Table 5, the third row)?

---

> ### Author Response · Authors · 2023-11-23
> **Response to Reviewer NVsT**
>
> We deeply appreciate your thorough and insightful reviews. We trust that our response adequately addresses any concerns you may have.
>
> **Q1. Our novelty and Clarify for Figure 2**
>
> You pointed out that Dense-JEPA's novelty is questionable, particularly in reference to Figure 2's similarity to I-JEPA's Figure 2. We acknowledge this observation and updated Figure 2 to better illustrate the distinctions between Dense-JEPA and I-JEPA.
>
> Regarding why our work is important and have significance, we believe that mask reconstruction on latent space will have gain much attention on its efficacy and efficiency, so study of training objective on latent space is crucial. Therefore, our investigation of the latent objective is invaluable and could inspire future research direction to explore better SSL objective, unlike the previous stereotype of masked reconstruction objective on pixel space.
> To clarify, Dense-JEPA's novel contribution lies in its unique approach to enhancing target representations. Unlike I-JEPA, Dense-JEPA employs a more nuanced strategy in selecting neighboring patches based on semantic similarity. This approach leads to a more accurate and granular understanding of local semantics, significantly contributing to the field of dense representation learning.
>
> **Q2. Clartiy for MSN and LAT**
>
> We agree that the descriptions of the MSN (Masked Semantic Neighboring) and LAT (Local Aggregation Target) modules could be more precise. MSN is indeed preparatory for LAT, but it is crucial for establishing fine-grained semantic relations among patches. On the other hand, LAT focuses on a coarser level of aggregation.
>
> In the revised paper, we provided a clearer distinction between these two modules in Figure 2. MSN is not just a preparatory step but a critical component in refining the target representation by identifying the most semantically relevant neighbors. LAT then aggregates these for a comprehensive representation. We will revise further for better clarity in the final draft.
>
> Regarding your question about the role of MSN without LAT (Table 5, second row), MSN independently contributes to the training procedure by identifying the most semantically relevant patches. This process is crucial for Dense-JEPA's effectiveness, as it ensures that even in the absence of LAT, the model doesn't revert to a coarse, generalized representation but maintains a degree of fine-grained semantic understanding (e.g., 53.7->55.2 scores in Table 5). Adding MSN to the vanilla baseline highly increases the results of the video segmentation performance regarding all metrics, which validates the benefit of masked semantic neighboring in finding semantically similar neighboring patches for masked patches (e.g., 53.7->55.2 scores in Table 5).
>
> Without MSN, Dense-JEPA would lack the nuanced understanding of local semantics, leading to a more generalized and potentially less accurate representation (e.g., 57.1->54.6 scores in Table 5).

---

> ### Author Response · Authors · 2023-11-23
> **Response to Reviewer NVsT (cont.)**
>
> Dear Reviewer NVsT,
>
> Thank you once again for your valuable efforts and constructive feedback on our paper. As the discussion period comes to a close, we trust that your concerns have been thoroughly addressed in our response. We dedicated significant effort to this, incorporating several new experiments, visualizations, and discussions. We sincerely hope you will consider our reply in your assessment. We look forward to hearing from you, and if the opportunity arises, we can further address unclear explanations and remaining concerns if any.
>
> Regards,
>
> Authors

---

### Author Response · Authors · 2023-11-23
**General Response**

Dear reviewers and AC,

We would like to express our gratitude for the opportunity to address the feedback from the reviewers on our submission regarding Dense-JEPA. We appreciate the insightful comments and constructive criticism offered by each reviewer, and we have taken careful steps to address the concerns raised. In accordance with your comments, we have carefully revised the manuscript with the following additional visualizations, discussions and experiments:

- (NVsT, wqKm, KGbU) Figure clarity (in Figure 2)
- (wqKm) Finetuning comparision with DINO, DINOv2 and MAE (in Appendix A)
- (NVsT, wqKm, RoAg, KGbU) Further clarification on our method (in Appendix B)
- (wqKm) Additional visualizations (in Appendix C)

We highlighted the revised contents in red (and the entire Appendix) for your convenience to check. We sincerely believe that Dense-JEPA can be a useful addition to the ICLR community, especially, as the revision allows us to better deliver the effectiveness of our method.

Thank you very much!

Authors.

---

### Meta-Review · Area_Chair_UX7v · 2023-12-11

**Metareview:**

This paper proposes a modification to the I-JEPA approach for self-supervised learning to learn better representations for dense tasks. The modification involves local aggregation and masked semantic neighbors. Experiments illustrate that the proposed approach does indeed lead to modest improvements in performance.

Although the paper does present promising initial results, some concerns were raised. Namely, several parts of the presentation were unclear in the original submission, especially in terms of details of the proposed methodology and how it is different from prior work. This was improved to some extent during the rebuttal. In addition, a more careful comparison to other previous work on dense SSL tasks, such as those noted by reviewer wqKm, would greatly strengthen the paper.

In the end, our opinion is that the weaknesses currently outweigh the strengths of the paper, hence the recommendation to reject.

**Justification For Why Not Higher Score:**

Several aspects of the proposed approach remain unclear, and these details likely matter for achieving the reported performance. In addition, the comparison to other prior approaches could be strengthened and made a more core part of the paper.

**Justification For Why Not Lower Score:**

N/A

---

### Decision · Program_Chairs · 2024-01-16

Reject